# Illumina RNA and SMRT Sequencing Reveals the Mechanism of Uptake and Transformation of Selenium Nanoparticles in Soybean Seedlings

**DOI:** 10.3390/plants12040789

**Published:** 2023-02-09

**Authors:** Yuzhou Xiong, Xumin Xiang, Chunmei Xiao, Na Zhang, Hua Cheng, Shen Rao, Shuiyuan Cheng, Li Li

**Affiliations:** School of Modern Industry for Selenium Science and Engineering, National R&D Center for Se-Rich Agricultural Products Processing Technology, Wuhan Polytechnic University, Wuhan 430023, China

**Keywords:** *Glycine max*, full-length transcriptome, SeNPs metabolism, WGCNA, lncRNA

## Abstract

Selenium (Se) is an essential element for mammals, and its deficiency in the diet is a global problem. Agronomic biofortification through exogenous Se provides a valuable strategy to enhance human Se intake. Selenium nanoparticles (SeNPs) have been regarded to be higher bioavailability and less toxicity in comparison with selenite and selenate. Still, little has been known about the mechanism of their metabolism in plants. Soybean (*Glycine max* L.) can enrich Se, providing an ideal carrier for Se biofortification. In this study, soybean sprouts were treated with SeNPs, and a combination of next-generation sequencing (NGS) and single-molecule real-time (SMRT) sequencing was applied to clarify the underlying molecular mechanism of SeNPs metabolism. A total of 74,662 nonredundant transcripts were obtained, and 2109 transcription factors, 9687 alternative splice events, and 3309 long non-coding RNAs (lncRNAs) were predicted, respectively. KEGG enrichment analysis of the DEGs revealed that metabolic pathways, biosynthesis of secondary metabolites, and peroxisome were most enriched both in roots and leaves after exposure to SeNPs. A total of 117 transcripts were identified to be putatively involved in SeNPs transport and biotransformation in soybean. The top six hub genes and their closely coexpressed Se metabolism-related genes, such as *adenylylsulfate reductase* (*APR3*), *methionine-tRNA ligase* (*SYM*), and *chloroplastic Nifs-like cysteine desulfurases* (*CNIF1*), were screened by WGCNA and identified to play crucial roles in SeNPs accumulation and tolerance in soybean. Finally, a putative metabolism pathway of SeNPs in soybean was proposed. These findings have provided a theoretical foundation for future elucidation of the mechanism of SeNPs metabolism in plants.

## 1. Introduction

Selenium (Se) is an essential trace mineral for humans and animals. It is beneficial to humans for its antioxidant, anticarcinogenic effects, and immune properties [1]. In humans, Se deficiency may cause some endemic diseases, such as Kashin–Beck disease and Keshan disease, and increase the incidence of various cancers and male infertility [2,3]. The United States Recommended Dietary Allowance for Se in adults is 55–75 µg/day. However, it is estimated that 15% (0.5–1.1 billion) of the global population, including people in parts of China, Siberia, Scandinavia, Sub-Saharan Africa, and New Zealand, is suffering from selenium deficiency due to very low levels of locally produced foods [4,5]. Se-enriched plants are the predominant source of dietary Se for consumers. Therefore, increasing the concentration of Se in edible plants’ parts/products is a viable pathway to overcome human Se deficiency [6,7,8]. Furthermore, it is important to note that organic forms of selenium are better at protecting the body against deficiencies of this element than inorganic forms [9].

Se seems not essential for plants, and excessive Se can be toxic due to changing structure and function of proteins and inducing oxidative/nitrosative stress [10,11,12]. Nevertheless, numerous reports indicate that Se in low concentrations has beneficial effects on plants, including growth promotion, improvement of antioxidative capacity, and resistance to abiotic and biotic stress [13,14]. The boundary between these two dual effects (beneficial and toxic) is very narrow and differs among plant species as well as exogenous Se forms applied [15,16]. Therefore, it is very important to further understand the uptake and metabolism of different Se species in plants. Current research has speculated on the transport and transformational processes of selenate and selenite in higher plant species [6,17,18]. Selenate is taken up via sulfate transporters (SULTRs) [19], and selenite is taken up by phosphorus (P) and silicon (Si) transporters [20,21]. By analog with sulfate, selenate, after absorption, can enter the sulfur reductive assimilation pathway. It can be activated by the enzyme ATP sulfurylase (APS) to form adenosine 5′-phosphosulfate/selenate (APS/APSe) and subsequently reduced by APS/APSe reductase (APR) to selenite [19]. This step mainly occurs in plastids and has been proven to be rate-limiting for Se assimilation in Indian Mustard [22]. Selenite may be further reduced enzymatically by sulfite reductase (SiR) or non-enzymatically by reduced glutathione (GSH) to produce selenide, which is incorporated into selenocysteine (SeCys) under the action of cysteine synthase (CS) complex [17]. SeCys is assumed to have poisonous effects on plants due to the substitution of Cys in participant proteins and leading to protein dysfunction [23]. It can be methylated by selenocysteine methyltransferase (SMT) to produce non-protein amino acid methyl-selenocysteine (MeSeCys) [24] or be converted to selenomethionine (SeMet) catalyzed by three enzymes: cystathionine gamma-synthase (CGS), cystathionine beta-lyase (METC), and 5-methyltetrahydropteroyltriglutamate--homocysteine methyltransferase (METE), where CGS is considered to be the rate-limiting enzyme for SeMet and volatile dimethylselenide (DMSe) production [25]. Selenocystathionine (SeCysTH) and selenohomocysteine (SeHCys) are the intermediates products in the conversion of SeCys to SeMet, which was confirmed by in vitro analysis of crude plant extracts [26,27]. Alternatively, SeCys may also be cleaved to elemental Se and alanine by chloroplastic cysteine desulfurase (CpNifS), whose function is to produce free S from Cys for the formation of Fe-S clusters and is important to the photosynthetic electron transport chain [28,29]. In *Arabidopsis*, *AtCpNIFS* encodes a PLP-dependent enzyme with SeCys lyase and Cys desulfurase activities, which contributes to selenium tolerance by preventing Se incorporation into protein [29]. 

In recent years, research on selenium nanoparticles (SeNPs) has gained more attention due to their important role in many physiological processes [30,31,32,33]. SeNPs have been demonstrated to possess anticancer and antimicrobial properties that may contribute to human health, not only as dietary supplements but also as therapeutic agents [34]. Due to their high area-to-volume ratio, plants can easily absorb SeNPs and possess more prominent bioactivity and biosafety properties than inorganic and organic Se [30]. In addition, SeNPs exhibited a slow-release effect in soil, which can promote the growth of soil microorganisms and enriched soil probiotics [35], providing a more environment-friendly Se additive for agriculture applications. Three methods can be employed for the synthesis of SeNPs: physical, chemical, and biological, among which microbial reduction from Se oxyanions is frequent in the environment [36]. Recently, several reports in plants have underlined the beneficial effects of SeNPs on modifying the expression of genes, antioxidant machinery, and primary and secondary metabolism [32,37,38,39]. However, the knowledge of their fate in plants and the mechanism therein is still scarce.

Soybean (*Glycine max* L.) is the most important food and oilseed crop, with a yearly production that exceeds 352 million tons planted on more than 123 million hectares [40]. Soybean contains high levels of protein. Reports indicate that the selenium taken up by crop grains mainly binds to proteins forming Se-containing proteins (SEPs), which supply the essential amino acids for human health requirements and possess physicochemical functionalities both of Se and proteins [41,42]. From a nutritional standpoint, soybean, as a good producer of SEPs, provides a high-quality material for food processing and an ideal carrier for Se biofortification [43]. 

In this study, we combined single-molecule real-time (SMRT) sequencing and next-generation sequencing (NGS) technology to get insight into the underlying mechanism of SeNPs metabolism in soybean. The involvement of alternative splicing and lncRNAs in SeNPs metabolism was also documented. The findings of this study will provide a valuable resource for understanding the molecular process of SeNPs transport and biotransformation in plants.

## 2. Results

### 2.1. Total Se Content and Se Species in Soybean

An overview of the experimental procedure is illustrated in Figure 1. The total Se content and species in soybean plants varied concerning the different Se treatments (Table 1). Compared with the control, the total Se content in leaves after application of 20 (L_N20) and 100 μmol/L SeNPs (L_N100) was remarkably increased by 9.71- and 28.66-fold, respectively, while the increment of Se content in roots was 68.47- and 161.98-fold, respectively. In addition, the total Se contents in roots were much higher than that in leaves in the same treatment, indicating that Se was mainly stored in soybean roots after SeNPs uptake from the culture solution. These results are consistent with the findings of Wang et al. [44] and Hu et al. [45], which suggested that SeNPs were prone to be transformed into organic Se species and concentrated in roots.

Only two Se species [Se^4+^ and SeMet] were observed in soybean tissue without Se supply. However, in the presence of SeNPs, there were five Se species [SeCys_2_, MeSeCys, SeMet, Se^4+^ and Se^6+^] detected in soybean tissues (Table 1). The main Se species in leaves was SeMet, which accounted for 55.14–70.89% of the total identifiable Se species under SeNPs treatment (Table 1). In roots, SeMet (33.13%) was still the major species when supplied with 20 μmol/L SeNPs (R_N20), but after the application of 100 μmol/L SeNPs (R_N100), MeSeCys (25.89%), Se^6+^ (24.62%), SeCys_2_ (22.64%) and SeMet (20.55%) were nearly equivalent in concentration. The concentration of Se^4+^ only accounted for approximately 2.30–2.14% (in leaves) and 6.30–7.54% (in roots) of total Se species content in SeNPs treatment (Table 1). Table 1 shows that the concentration of different Se species also increased with the Se level added to the hydropic system.

### 2.2. Sequencing Data Statistical Analysis of SMRT and Illumina RNA-Seq

To identify and characterize the transcriptome of soybean under SeNPs exposure, we measured the roots and leaves under different SeNPs treatments (L_CK, L_N20, L_N100, R_CK, R_N20, R_N100) by combining the PacBio SMRT and NGS technologies for whole-transcriptome profiling. In total, 48.87 Gb of raw reads was generated using the PacBio Sequel II System, which resulted in 46.04 Gb of post-filter subreads (length > 50 bp and accuracy > 0.75). A total of 773,038 reads of inserts (ROIs) (average read length of 1682 bp), of which 627,372 (81.16% of the total reads) were full-length non-chimeric (FLNC) reads, were extracted from the original sequences. The FLNC reads were clustered using the ICE program and polished using the SMRT analysis software (version 2.3), and the low-quality isoforms were then corrected via Illumina RNA-seq data. We finally obtained 74,662 polished sequences with an average length of 1486 bp and N50 of 1943 bp. Most of the polished consensus sequences (98.95%) were mapped to the reference genome (*G*. *max* v4.0) (https://ftp.ncbi.nlm.nih.gov/genomes/all/GCF/000/004/515, accessed on 1 December 2022) using GMAP software (version 2021-12-17) [46] (Appendix A). Further classification of the mapped reads revealed that 12,256 reads were multiple mapped; 32,699 and 28,921 were mapped to the positive strand (+) and opposite strand (−) of the genome, respectively (Appendix A). Next, we compared the 42,215 corrected transcripts against the soybean genome set (*G. max* v4.0). The results showed that 22,244 (52.69%) novel isoforms came from annotated genes and 2534 (6.00%) novel isoforms did not overlap with any annotated genes (Appendix A). To study the annotation information of the new genes, we conducted BLAST searches in seven databases (NR, NT, Pfam, KOG/COG, SwissProt, KEGG, GO). In total, 445 of these novel genes were successfully annotated in all seven databases, and 1813 isoforms were successfully annotated by at least one database (Appendix A).

### 2.3. Analysis of DEGs in Response to Different SeNPs Treatment

In order to study the gene expression response to SeNPs, significant DEGs were obtained from the roots and leaves of SeNPs-treated plants compared to the control group, with the filter criteria of fold change set at ≥2.0 and FDR at ≤0.05. As a result, 393 (44 up- and 349 down-regulated), 897 (172 up- and 725 down-regulated), 719 (458 up- and 261 down-regulated), and 5855 (2807 up- and 3048 down-regulated) DEGs were obtained from L_N20 vs. L_CK, L_N100 vs. L_CK, R_N20 vs. R_CK, R_N100 vs. R_CK, respectively (Figure 2a). SeNPs at 100 μmol/L induced more extensive changes in gene expression compared to 20 μmol/L, and the genes in root tissues showed more sensitivity than those in leaves exposed to the same exogenous Se level (Figure 2a). At the same time, more genes were down-regulated than up-regulated, especially after 100 μmol/L treatment (Figure 2a). This may be due to the inhibitory effects on soybean seedlings when exposed to high Se [47]. As shown in the Venn diagram (Figure 2b), 16 DEGs in both roots and leaves were commonly detected in response to SeNPs treatments. Interestingly, among them, eleven genes were associated with circadian rhythms, including five transcription factors (LOC100101861, LOC778089, MYB114, LCL3, and LCL1), three transcriptional cofactors (LOC100800977, LOC100779717, and LOC100781565), and three UV receptor proteins (LOC100778841, LOC100781488, and LOC100807837) which can be activated by transcription of responsive clock genes. 

We further analyzed the functional roles of DEGs between the comparison groups. The top GO classifications in biological process (BP), molecular function (MF), and cellular component (CC) are displayed in Appendix A. The top three enriched MFs were oxidoreductase activity (GO:0016491; 57 genes), iron ion binding (GO:0005506; 19 genes), and protein disulfide oxidoreductase activity (GO:0015035; 8 genes) for L_N20 vs. L_CK, catalytic activity (GO:0003824; 385 genes), oxidoreductase activity (GO:0016491; 119 genes) and cofactor binding (GO:0003824; 44 genes) for L_N100 vs. L_CK, catalytic activity (GO:0003824; 344 genes), oxidoreductase activity (GO:0016491; 95 genes) and iron ion binding (GO:0005506; 23 genes) for R_N20 vs. R_CK, catalytic activity (GO:0003824; 2495 genes), ion binding (GO:0043167; 1501 genes) and oxidoreductase activity (GO:0016491; 664 genes) for R_N100 vs. R_CK, respectively. The top two enriched BPs were the oxidation-reduction process (GO:0055114; 47 genes) and lipid metabolic process (GO:0006629; 28 genes) for L_N20 vs. L_CK, single-organism process (GO:0044699; 311 genes) and single-organism metabolic process (GO:0044710; 182 genes) for L_N100 vs. L_CK, single-organism process (GO:0044699; 248 genes) and oxidation-reduction process (GO:0055114; 99 genes) for R_N20 vs. R_CK, metabolic process (GO:0008152; 2623 genes) and single-organism process (GO:0044699; 1934 genes) for R_N100 vs. R_CK, respectively.

The most obviously enriched metabolic pathway (q-value ≤ 0.05) in L_N20 vs. L_CK, L_N100 vs. L_CK, R_N20 vs. R_CK, and R_N100 vs. R_CK, totaled in 4, 22, 18 and 40, respectively. The 20 most significant KEGG pathways are displayed in Figure 2c and Appendix A. The top enriched pathways for all comparison groups were metabolic pathways, biosynthesis of secondary metabolites, and peroxisome, suggesting these vital activities were remarkably affected by the treatment of SeNPs in soybean (Figure 2c, Appendix A). The phenylpropanoid biosynthesis, phenylalanine metabolism, taurine and hypotaurine metabolism, cysteine, and methionine metabolism, flavonoid biosynthesis, plant-pathogen interaction, glutathione metabolism, and starch and sucrose metabolism were uniquely significantly enriched in root comparison groups. The last four pathways only enriched in R_N100 vs. R_CK and R_N20 vs. R_N100, indicating which may play an important role in the regulation of tolerance of SeNPs. In leaves, carotenoid biosynthesis and porphyrin and chlorophyll metabolism were the only most common enriched metabolic pathways in L_N20 vs. L_CK and L_N100 vs. L_CK (Figure 2c), which is in line with our previous physiological results [48]. 

### 2.4. Characterization of the Genes Involved in Se Metabolism

Transporters for S, P, Si, Amino acid (AA), and aquaporins were screened, which may be involved in the uptake and translocation of Se in plants. In total, 66 transporter genes: six members of *SULTRs*, including *SULTR 1;3* (2 isoforms), *SULTR 2;1* (3 isoforms), *SULTR 3;1* (4 isoforms), *SULTR 3;3* (2 isoforms), *SULTR 3;4* (3 isoforms) and *SULTR 4;1* (2 isoforms), three members of phosphorus transporters, including *PHO1* (7 isoforms), *PHT1* (9 isoforms) and *PHT2* (3 isoforms), three members of high-affinity nitrate transporters, including *NRT2;4* (3 isoforms), *NRT2;5* (2 isoforms) and *NRT3;1*(2 isoforms), one member of amino acid permease (*APP6*, 2 isoforms), 17 isoforms of aquaporin, including 2 nodulin26-like intrinsic proteins (*NIPs*), 10 plasma membrane intrinsic proteins (*PIPs*) and 5 tonoplast intrinsic proteins (*TIPs*), and 5 isoforms of the silicon efflux transporter (*Lsi2-like*) were identified (Appendix A). At the same time, 45 assimilatory enzyme genes: *ATP sulfurylase* (*APS*, 3 isoforms), *5’-adenylylsulfate reductase* (*APR*, 3 isoforms), *thioredoxin reductase* (*TRXB*, 5 isoforms), *cysteine synthase* (*CYSK*, 2 isoforms), *selenocysteine methyltransferase* (*SMT*, 1 isoform), *cysteine desulfurases in chloroplastic* (*CNIF*, 5 isoforms), *cysteine desulfurase in mitochondrial* (*MNIF*, 2 isoforms), *probably cysteine desulfurase* (*CSD*, 1 isoform), *cystathionine gamma-synthase* (*CGS*, 2 isoforms), *cystathionine beta-lyase* (*METC*, 2 isoforms), *homocysteine methyltransferase* (*METE*, 5 isoforms), *methionine S-methyltransferase* (*MMT*, 3 isoforms), *methionine-tRNA ligase* (*SYM*, 6 isoforms), *cysteine-tRNA ligase* (*SYC*, 4 isoforms) and *methionine gamma-lyase* (*MGL*, 1 isoform) were identified in our study (Appendix A). In plants, selenoproteins act in the transport of Se and regulation of cellular redox balance. Here, six genes containing *selenium binding protein* (*SBP*, three isoforms), *selenoprotein H*, and *selenoprotein F* (*SEP15*, two isoforms) were identified as specific genes which may be related to Se metabolism (Appendix A). 

The expression profile analysis of these 117 transcripts from the six treatment sample groups is shown in Figure 3. SeNPs, especially in the 100 μmol/L treatment, caused significant variations in the mRNA levels of sulfate transporters (SULTR 1;3, SULTR 2;1, SULTR 3;1), phosphate transporters (PHO1; H7, PHO1; H8, PHO1; H9), silicon transporters (LSI2.1), aquaporins (PIP2;2, PIP2;5, PIP2;7, NIP5;1, NIP6;1, TIP4;1) and nitrate transporters (NRT2;4, NRT2;5, NRT3;1). Unlike *sulfate transporter 1;3*, which was substantially induced by SeNPs both in roots and leaves, the expression of *SULTR 2;1* and *SULTR 3;1* was up-regulated in leaves but significantly down-regulated in roots after the application of SeNPs. In addition, 17 transcripts involved in Se biotransformation were differentially expressed by SeNPs treatments (Appendix A), including *APS2*, *APR3*, *MGL*, *CGS1*, *CSD*, *SYM*, *MMT1*, *METE*, *CYSK-1*, *CYSK-2*, *CNIF3-1*, *CNIF3-2*, *CNIF3-3*, *TRXB3-1*, *TRXB3-2*, and *TRXB2-1*, *TRXB2-2*. SeNPs exposure increased the FPKM of *CYSK-1*, *SMT*, *MMT*, *CNIF3-1*, *SBP1*, *SYM*, and *SYC* but reduced those of *APS1*, *APS2*, *APR3*, *CGS1*, *METE*, *MGL*, *MMT1*, *CSD*, *CNIF3-2* and *MNIF1* in soybean roots and/or in leaves.

### 2.5. Identification of Transcription Factors (TFs), Alternative Splicing (AS) Events and lncRNAs 

TFs are the key to the regulation of gene expression. This study predicted 2109 TFs categorized into 29 TF families from the nonredundant transcripts (Figure 4a). The most abundant TF family was the WRKY (221 genes, 10.48%) family, followed by the bHLH (151 genes, 7.16%), NAC (138 genes, 6.54%), AP2/ERF-ERF (133 genes, 6.31%), GRAS (116 genes, 5.50%), and AUX/IAA (115 genes, 5.45%). Several common TFs, such as C2H2 and MYB, were also included in the top 15 TF families.

The AS events of soybean seedlings treated with SeNPs were predicted from the nonredundant transcripts. In total, 9687 AS events were detected, which were classified into seven categories: alternative 5′ donor (A5), alternative 3′ donor (A3), alternative first exon (AF), alternative last exon (AL), retention intron (RI), mutually exclusive exons (MX) and exon skipping (SE) events. In this study, A3 was the major type of AS event, which accounted for 31.23% (3025), followed by RI (2087, 21.55%), A5 (1771, 18.29%), SE (1653, 17.07%), and AF (875, 9.03%) (Figure 4b). The AL and MX types of AS were found in only 243 (2.51%) and 31 (0.32%) isoforms, respectively. At the same time, 26 Se metabolism-related genes were identified with splicing changes. Among these, eight genes, including seven AS isoforms of CGS, four AS isoforms of LOC100804208 (SULTR 1;3), three AS isoforms of LOC100819254 (TRXB3) and LOC100803724 (MMT1), and two AS isoforms of LOC100791198 (Lsi2), LOC100794958 (CYSK), LOC100819536 (APR3) and LOC100794561 (TRXB3), showed significantly different expression after SeNPs application. These results suggested that AS may affect protein-coding sequences or generate non-productive mRNAs to influence transcription levels in selenium metabolism in soybean. 

LncRNAs are known to play important roles in plant development and stress response. A total of 7006, 12,892, 3711, and 5390 lncRNA candidates were evaluated using the databases CNCI, Pfam, PLEK, and CPC, accordingly (Figure 4c). Finally, 3309 high-confidence lncRNAs were identified and further classified into four categories, i.e., sense intronic-lncRNA (63.86%), sense overlapping-lncRNA (24.21%), antisense-lncRNA (8.79%), and lincRNA (3.14%) according to their genomic origins (Figure 4d). In total, 491 differentially expressed lncRNAs (DE-lncRNAs) were obtained in response to SeNPs treatment. More precisely, 18 and 64 in leaves, as well as 49 and 424 DE-lncRNAs in roots, were identified under the application of 20 and 100 μmol/L SeNPs, respectively (Figure 4e). Moreover, in roots, the numbers of up-regulated DE-lncRNAs were greater than those of down-regulated; in contrast, more down-regulated DE-lncRNAs were found in SeNPs-treated leaves. Relative expression levels of five randomly selected DE-lncRNAs were further verified using RT-qPCR (Appendix A). The correlation analysis results revealed that our RNA-seq data are highly reproducible and reliable and can be used for further study.

### 2.6. WGCNA Analysis of TFs and DE-lncRNAs Related to Se Metabolism in Soybean Supplied with SeNPs

To explore the potential regulatory function of TFs and lncRNAs associated with Se-metabolism, WGCNA was performed to analyze the coexpression relationship among the 1435 TFs (the transcripts with normalized reads lower than 1.0 FPKM were removed from 2109 TFs), 491 DE-lncRNAs and 117 transcripts involved in Se transport and assimilation genes. We successfully obtained six modules: MEbrown (287 genes), MEturquoise (790 genes), MEyellow (227 genes), MEblue (616 genes), MEgreen (34 genes), and MEgrey (25 genes). The MEblue module was found to be significantly correlated with contents of SeCys_2_, MeSeCys, Se^4+^, Se^6+^, and total Se (Pearson’s correlation coefficient > 0.90 and *p* ≤ 0.01) (Figure 5a) and may play a key role in the metabolism of SeNPs in soybean. Genes in the MEblue module were then analyzed, and 20 hub genes were identified, including 13 TFs and 7 DE-lncRNAs (Figure 5b). Among these, three DE-lncRNAs (*LOC100795325*, *LOC100527660*, *LOC100790467*) and three TFs (*LOC100500420*, *LOC100794643*, *LOC100816770*) were the top six genes with the highest connectivity. Additionally, interaction patterns between lncRNAs/TFs and structural genes related to Se metabolism were also recognized within the MEblue network (with weight > 0.9) (Figure 5c). The results showed that *SYM* (*LOC100786507*), *APR3* (*LOC100799978*), and *CNIF3* (*LOC100796732*) coexpressed closely with twelve genes (including the top six genes) in different regulation models. For example, *SYM* was regulated by *LOC10079525*, *LOC100527660*, *LOC100500420*, and *LOC100816770*; while DE-lncRNAs *LOC100795325*, *LOC100527660*, and TF gene *LOC100500420* have the potential to regulate all of *SYM*, *APR3,* and *CNIF3*, revealing these DE-lncRNAs and TFs may affect the expression of targeting the gene in a complex way and serve as the important regulator of gene coexpression networks. Furthermore, GO and KEGG enrichment analyses of TFs and structural genes related to Se metabolism were also conducted for the MEblue modules. The GO analysis results showed that MEblue module genes were involved mainly in “response to hormone”, “phosphorelay signal transduction system”, “N-acetyltransferase activity”, “FtsZ-dependent cytokinesis”, “division septum assembly”, “kinetochore assembly”, “histone binding” and “protein trimerization” (Appendix A). KEGG enrichment analysis showed that “Plant hormone signal transduction”, “MAPK signaling pathway-plant”, “circadian rhythm-plant” and “sulfur metabolism” were the most abundant pathway in the MEblue modules (Appendix A).

## 3. Discussion

Fortification of Se is of vital importance for both nutritional demand and prevention of Se-deficiency-related diseases. Different crops exhibit the varying ability to enrich Se, with beans performing much better than other cereals. In this study, soybean seedlings showed a strong ability to accumulate Se when treated with 100 μmol/L SeNPs; the total Se concentration in the leaves of seedlings reached 42.85 mg/kg, and those in the roots reached 518.01 mg/kg. Therefore, soybean sprouts as dietary selenium supplements have a broad market. 

Plants take up both inorganic (selenate, selenite, and nano-sized elemental Se) and organic (e.g., Se-amino acids) selenium species. However, different Se species can be absorbed and translocated via different mechanisms [49]. It is well established that the absorption of selenate and selenite is through high-affinity sulfate transporters (SULTR1;1 and SULTR1;2) and phosphate (PHT1;1, PHT1;2, PHT1;3, and PHT1;4) and silicon transporters (LSI1/NIP2;1), respectively, and SeMet through NRT 1.1B [27]. However, less is known about the mechanisms of plant uptake of selenium nanoparticles; only a previous study reported that the wheat root absorption of SeNPs could be inhibited by the aquaporins inhibitor AgNO_3_ by above 90% [45]. Aquaporins play important roles in the transport of water and substrates in plants and in responses to drought, cold, or salt stresses [50]. The influx of metalloids into the root cells was reported to be somewhat similarly mediated by nodulin26-like intrinsic protein (NIP) members, and some of them shared similar pathways for B, Si, and As [51]. Se is a metalloid element with similar chemical properties to B, Si, and As. In this study, we found that a total of 17 aquaporin genes (belonging to three subfamilies: PIPs, TIPs, and NIPs) were differentially expressed in the SeNPs-treated soybean seedlings. Among these, *NIP5;1* and *NIP6;1* mRNA levels in roots were significantly down-regulated when treated with 100 μmol/L SeNPs. In *Arabidopsis*, *AtNIP5;1* and *AtNIP6;1* encode plasma membrane-localized transporters for B(OH)_3_: *AtNIP5;1* is mainly involved in the uptake from soils, while *AtNIP6;1* plays an important role in the intra-vascular transfer of B to the developing tissues [51]. The decreased expression of *NIP5;1* and *NIP6;1* in soybean roots in response to high Se may contribute to minimizing excess Se entry to the root cells. 

In this study, we also observed two *SULTR2;1*, one *SULTR1;3*, three *SULTR 3;1* and three *PHO1s* (*GmPHO1;H7*, *GmPHO1;H8*, *GmPHO1;H9*) genes differentially expressed after SeNPs application. Previous works have demonstrated that *AtSULTR2;1* and *AtSULTR1;3* were separately located in xylem parenchyma cells and phloem companion cells in *Arabidopsis* and participated in sulfate transport from source to sink organs, while *SULTR3;1* was expressed in chloroplast and involved in sulfate uptake across the chloroplast/plastid envelope membrane [52,53]. Takahashi et al. [54] observed that the expression of SULTR2;1 was activated in roots while strongly repressed in shoots by sulfur starvation, which facilitates the distribution of sulfate to leaf tissues under sulfur-limited conditions. Our results also displayed contrasting patterns of *SULTR2;1* mRNA accumulation between roots and shoots in soybean, with a significant decrease in roots and increase in shoots when supplied with SeNPs. *AtPHO1* is predominantly expressed in root stelar cells and participates in loading inorganic Pi into the xylem of roots in *Arabidopsis* [55]. The transgenic plants harboring *GmPHO1;H8,* and *GmPHO1; H9* showed elevated sensitivity to Pi-deficiency and strong tolerance to salinity stress [56]. In addition, Cao et al. [57] found that the expression of *CsPHO1;2* exhibited significantly positive correlations with the accumulation of Se, and they speculated this gene might play a role in MeSeCys transport and accumulation in aerial parts of tea plants. Our observations lend support to hypothesize that promotion of *SULTR1;3* in both shoots and roots, as well as induction of *SULTR2;1*, *SULTR3;1* and *PHO1s* in leaves vs. suppression in roots, contributes to long-distance transport of Se nutrients to the sink organs via phloem and blocking the root-to-shoot transport via the xylem, and then retaining Se in the roots. It is, therefore, reasonable to conclude that the high content of Se in roots may be attributed to the action of these transporters, which may also play key roles in maintaining Se homeostasis between cell membranes and organs in soybean plants under high SeNPs conditions.

Previous studies demonstrated that the transformation of selenocompounds varies with different plant species and different exogenous Se forms. The Allium and cruciferous vegetables mainly accumulated MeSeCys and SeMet, while most grains accumulated SeMet [58]. Funes-Collado et al. [59] analyzed the changes in Se speciation of three leguminous sprouts (alfalfa, soy, and lentil) supplied with different Se-enriched solutions (mainly in selenite and selenate). They found that during the growth period, part of the inorganic Se was transformed into SeCys_2_, and a larger proportion was transformed into SeMet. In another experiment, SeMet (41.5–80.5%) and MeSeCys (19.5–21.2%) were the dominant Se forms in all selenite-fortified soybean sprouts, while SeCys_2_ was only found when Se was applied at the above 30 mg/L [60]. Here, five Se species, including SeCys_2_, MeSeCys, SeMet, Se^4+^, and Se^6+^, were detected in soybean roots and shoots (Table 1), suggesting that the absorbed SeNPs were converted to oxidative and organic Se species in roots. Some of them were then translocated and redistributed in other organs mediated by different transporters in former contexts. Such a phenomenon has been reported in *Triticum aestivum* [45], *Oryza sativa* [44], and *Allium sativum* [16] as well. Unlike in rice [44] and garlic [16], where no Se^6+^ was detected, in our study, apart from a small amount of Se^4+^, more Se^6+^ was detected in SeNPs exposed soybean tissues. It has been reported that selenite may be oxidized as sulfite by the catalysis of sulfite oxidase. Still, there is no evidence of how elemental Se can be converted to oxidative species (selenate or selenite) in the plant. The SeNPs assimilation mechanism warrants further studies using genetics and physiology methods.

After translocation and redistribution, the reduction of oxidative Se species occurs through a series of consecutive steps that convert selenate > selenite > selenide > SeCys. APS and APR play a key role in selenate assimilation, and APR is considered the rate-limiting enzyme [22]. A mutant *Arabidopsis* line (*apr2-1*) showed a high concentration of selenate and negligible amounts of selenite, and low S flux from sulfate to reduced S compounds such as GSH and proteins [61]. In addition, APR seems to be more rate-limiting for Se assimilation in non-hyperaccumulators than in hyperaccumulators [28]. In the present study, the expression of *APS2* and *APR3* were down-regulated significantly by SeNPs addition. Selenite is considered to reduce to selenide by enzyme sulfite reductase, thioredoxin system, or via glutathione-mediated reduction [17,62]. Meanwhile, SBP1 is also reported to reduce SeO_3_^2–^ to form an R–S–Se–S–R-type complex, which can circumvent reduction by GSH and thus prevent Se from interfering with sulfur metabolism [63]. Five *TRXB* members and three *SBP* members were identified in our study, but to our interest, *SiR* was not found in our data. In addition, we found the FPKM values of *SBP1*, *TRXB3*, and *GST* were significantly higher in SeNPs-exposed soybean roots. According to the results, it can be speculated that the reduction of selenite in soybean is mainly via interaction with glutathione or by the action of selenium-binding protein and thioredoxin reductase other than sulfite reductase. Our findings are in line with the report of Fish et al. [64] that the knockdown of *SiR* in *Arabidopsis* did not affect selenite reduction to selenide. In our data, the relatively higher content and percentage of Se^6+^ compared with lower Se^4+^ detected in soybean leaves and roots, even under high Se application (Table 1), could be attributed not only to the rapid consumption of selenite in the downstream reactions but also the decreased transcript levels of APS and APR. 

SeCys is synthesized by the donation of a carbon skeleton from O-acetylserine to selenide in the presence of a cysteine synthase enzyme. It can be transformed into elementary Se, MeSeCys, and SeMet, the latter being responsible for the origin of selenoproteins [18]. In this study, SeMet was the only organic species detected in non-Se treated soybean seedlings; after SeNPs were applied, three organic Se species, MeSeCys, SeCys_2,_ and SeMet, were all detected. In addition, the proportion of SeMet decreased from 70.89% to 55.14% in leaves, and from 33.13% to 20.55% in roots, as the exogenous Se rate developed from 20 to 100 μmol/L. At the same time, the proportion of SeCys_2_ and MeSeCys were both greatly elevated (Table 1). Consistently, we found that the transcript levels of corresponding genes, such as *CYSK* for Cys synthesis and *SMT* for MeSeCys synthesis, were all increased significantly. In contrast, *CGS* and *METE* for SeMet synthesis were all decreased by SeNPs (especially 100 μmol/L) addition.

Se toxicity in plants was explained partially to be driven by the unspecific SeCys incorporation into proteins [65], and the production of MeSeCys, volatile compounds and elemental Se were all reported to prevent the incorporation of SeCys in proteins and develop plant tolerance to Se [27]. In soybean, two isoforms of *CNIF3*, and two other genes, *MGL* and *MMT*, whose encoding products separately catalyze MeSeCys and SeMet to form a volatile compound methyl-seleno [66] and SeMet to produce DMSe [67], exhibited sharply down-regulated expression levels under 100 μmol/L SeNPs treatment compared to the control (Figure 3; Appendix A). Thus, our findings imply that soybean may have other detoxication pathways to cope with high exogenous selenium. SeMet and SeCys can be ligated onto tRNA and misincorporated at Met and Cys codons during protein synthesis. In our data, the expression of gene coding methionine-tRNA ligase (SYM, cytoplasmic) was significantly up-regulated by SeNPs treatment. In contrast, the transcripts of cysteine-tRNA ligase (SYC) showed a non-significant difference. Unlike selenocysteine, which is more reactive, selenomethione in proteins is not considered deleterious. The remarkable increase in the transcripts level of the *SYM* gene in SeNPs-exposed plants may produce more SeMet- and Met-contained proteins and then improve the nutritional values of soybean.

The present study was aimed at understanding the central players in SeNPs metabolism in soybean through an analysis of RNA-seq datasets. AS, TFs and lncRNAs are an extensive transcriptional and post-transcriptional regulatory mechanism that play crucial roles in diverse biological processes [68]. A previous study in rice has shown that AS event of the sulfate transporter OsSTLTR1;1 plays an important role in regulating the cellular sulfur status required for growth during stress conditions [69]. From Iso-Seq data, we discovered twenty-six Se metabolism-related genes were alternatively spliced, among which eight genes (2, 2, 7, 3, 2, 3, 4, 2 AS isoforms from *APR3*, *CYSK*, *CGS*, *TRXB3-1*, *TRXB3-2*, *MMT1*, *SULTR* 1;3 and *LSI2*, respectively) expressed differently after SeNPs application. These findings allow us to better understand the post-transcriptional regulatory mechanisms operating to regulate selenium uptake, transport, and assimilation by the plant. 

WGCNA is a systematic biological method that has been proven to be an efficient data mining method for conducting modular classification to determine coexpression modules with high biological significance [70]. In this paper, 1435 TFs, 491 DE-lncRNAs, and 117 Se metabolism-related transcripts were clustered into 6 gene modules, among which the MEblue module was most correlated with contents of SeCys_2_, MeSeCys, Se^4+^, Se^6+^, and total Se (Figure 5a). The MEblue module significantly enriched a large number of Se metabolism-related genes, such as Se assimilatory enzyme genes *APR3*, *CYSK*, *CGS1*, *SBP1*, *SMT*, *CNIF3*, *SYMC*, and aquaporin gene *PIP2;5*, sulfate transporter genes *SULTR1;3*, *SULTR3;4*, *SULTR4;1*, phosphorus transporter genes *PHT1;3*, *PHT1;7*, *PHT1;9*, *ABCC3*, *ABCC9* (Appendix A), indicating that these genes may play important roles in the transformation and transporting of SeNPs/selenocompound. Additionally, three DE-lncRNAs and three TFs were identified as the top six hub genes in the MEblue module. These hub genes are highly correlated with *SYM*, *APR3*, and *CNIF3*, which are important enzyme genes in selenocompound synthesis. Little is known about lncRNA related to Se metabolism in plants. Rao et al. [71] identified three lncRNAs with their targeted Se-related gene *APS1*, *APR3*, *CNIF1*, and *SAT2* in selenium hyperaccumulator *Cardamine violifolia* under selenate treatment, which was in line with our findings. Whether and how these lncRNAs and TFs interact with Se metabolism-related genes remains to be elucidated.

Finally, based on the general Selecompound biosynthesis pathway suggested in previous studies and the results obtained in the current study, we propose a putative metabolism pathway of SeNPs in soybean (Figure 6). In the diagram, several candidate genes for some key steps and their regulatory ways are suggested. In our opinion, they should be considered first for further functional studies. 

## 4. Materials and Methods

### 4.1. Plant Materials, Growth Conditions, and SeNPs Treatment

Soybean (*G. max*) cultivar “Dongnong 690” seeds provided by Wuhan Fengyuan Seed Co. Ltd. (Wuhan, China) were soaked in deionized water for 8 h, subsequently germinated on wet filter papers for 48 h. After that, the seeds were cultivated in a plastic hydroponic box (34 cm length × 25 cm width × 4.5 cm depth) under controlled conditions (24 °C, 16h photoperiod, and 75% relative humidity). The plastic hydroponic box was first added with 1L of deionized water, then replaced with SeNP solutions when the plantule reached a length of 1 cm. Bioactive SeNPs (particle size of 100–600 nm) were obtained from the Institute of Agricultural Economics and Technology, Hubei Academy of Agricultural Sciences (Wuhan, China). In our pre-experiments, the total Se content in soybean seedlings was increased with Se concentrations in nutrient solutions varying from 0, 10, 20, 40, 80 to 100 μM; when the Se level was 20 μM, soybean plants grew best, while Se at 100 μM showed significant inhibition for the plant growth. Therefore, the concentrations of SeNPs used in the experiment were set as 0 (L_CK, R_CK 20), 20 (L_N20, R_N20), and 100 μM (L_N100, R_N100). After 7 days of treatment, whole seedlings were harvested, washed thoroughly with deionized water, weighed, and separated into roots and leaves. Fresh samples were frozen in liquid nitrogen immediately and stored at −80 °C. Samples for Se detection were dried at 60 °C, crushed, and kept in an airtight container.

### 4.2. Determination of Total Se Content and Se Species

The total Se content was determined according to the method of Rao et al. [71]. In brief, 0.1 g of dry samples were digested using 7 mL of concentrated HNO_3_ (GR) in a microwave oven digestion system until they were clear. The clear solution was mixed with 10% HCl to 10 mL for total Se determination using a hydride generation atomic fluorescence spectrometry (HG—AFS) (AF8530, Beijing Haiguang Instrument Co., Beijing, China). The HG-AFS conditions were as follows: negative high voltage, 340 V; lamp current, 90 mA; atomization temperature, 800 °C; carrier gas flow rate, 500 mL/min; injection volume 1 mL.

Determination of Se speciation in soybean seedlings was performed using liquid chromatography-HG-AFS (LC-AFS530, Beijing Haiguang Instrument, Beijing, China) following the method of Yang et al. [58]. Then, 0.1 g of dry soybean tissue samples were enzymatically hydrolyzed with protease K and pronase E under ultrasonication at 37 °C for 1 h. The solution was then centrifuged at 10,000 rpm for 20 min. Finally, the supernatant was filtered through 0.22 μm cellulose nitrate filters for subsequent Se species analysis. The LC-HG-AFS conditions were as follows: chromatographic column, Hamilton PRP-X100 (Hamilton Co., Reno, NV, USA); mobile phase, 40 mmol/L KH_2_PO_4_ + 20 mmol/L KCl, pH 6.0; flow rate, 1.0 mL/min; column temperature, 27 °C; injection volume, 150 μL; negative high voltage, 350 V; cathodic current, 80 mA; carrier gas flow rate, 600 mL/min. Five standard selenocompounds: Se^4+^, Se^6+^, SeCys_2_, MeSeCys, and SeMet, were purchased from the National Research Center for Certified Reference Materials, Beijing, China. Experiments were carried out in three separate biological triplicates, each with three technical duplicates of Se determination.

### 4.3. RNA Isolation, Library Preparation, and Sequencing

The samples’ total RNA was isolated using the TaKaRa Plant RNA extraction kit (Takara, Japan) following the provided protocol, and genomic DNA was removed via digestion with DNase I (Takara, Japan). The library was prepared after the samples passed the quality test (OD260/280 values of 1.9 to 2.2, OD260/230 values ≥2.0, and RNA integrity number values > 6.8). A total of 18 cDNA libraries of six samples (L_CK, L_N20, L_N100, R_CK, R_N20, R_N100) with three repeats were constructed using the NEBNext^®^ UltraTm RNA Library Prep Kit (NEB, Ipswich, MA, USA) and sequenced on an Illumina HiSeq 2500 platform at Biomarker Tech. Co. (Beijing, China). Clean reads were obtained by removal of reads containing adapter, ploy-N, and low-quality reads from raw data. The remaining clean reads were then mapped to the PacBio reference sequence. The mapped fragments for each gene were counted, and fragments per kilobase per million mapped reads (FPKM) were calculated as the gene expression abundance. Differential expression analysis between two samples was performed using the DESeq R package (version 1.18.0). Transcripts with FDR ≤ 0.05 and |log2(fold change)| ≥ 1 were screened as DEGs. 

The same samples previously used for RNA-seq libraries were equally mixed and combined to generate one pool for PacBio full-length sequencing. FL cDNA was synthesized using the Clontech SMARTer PCR cDNA Synthesis Kit (Mountain View, CA, USA) and filtered using the BluePippinTm Size Selection System (Sage Science Beverly, MA, USA). One single-molecular real-time library of different lengths was constructed and then sequenced using PacBio Sequel II System at Novogene Bioinformatics Technology Co., Ltd. (Beijing, China). Raw reads were processed into error-corrected reads of inserts (ROIs) by the Iso-seq pipeline, with full passes > 0 and the accuracy of the sequence > 0.75. High-quality clean data were obtained by removing reads containing connectors and low-quality reads. Full-length non-chimeric reads (FLNCs) were then determined by searching for poly-A tail signals and the 5′ and 3′ cDNA primers in ROIs. Iterative clustering for error correction (ICE) was used to obtain consensus isoforms. The full-length consensus sequences from ICE were polished using SMRT analysis version 10.1 (www.pacb.com/support/software-downloads/, accessed on 1 December 2022). High-quality FL transcripts with a post-correction accuracy greater than 99% were generated for further analysis. Additional nucleotide errors in consensus reads were corrected using the Illumina RNA-seq data with the software LoRDEC (version 0.7) [72]. The error-corrected isoforms were mapped to the *G. max* v4.0 reference genome (https://ftp.ncbi.nlm.nih.gov/genomes/all/GCF/000/004/515, accessed on 4 October 2022) with GMAP software (version 2021-12-17) [46] and then the redundant isoforms (identity > 0.9, coverage < 0.85) were eliminated using the ToFU package (version 1.5.0). The resulting transcript sequence was directly used for subsequent analysis of AS events and lncRNAs.

### 4.4. Gene Functional Annotation and Identification of Transcripts Related to Se Metabolism

To obtain comprehensive annotation information, the nonredundant transcripts were aligned with NCBI NR, NT, Swiss Prot [73], GO [74], KOG/COG [75], Pfam [76], and KEGG databases by BLAST software (version 2.2.26) [77]. The GOseq R package (version 1.10.0) based on Wallenius noncentral hypergeometric distribution and KOBAS software (version 3.0) was subsequently used to analyze GO and KEGG enrichment, respectively. Furthermore, transcripts involved in Se metabolism were screened by searching the integrative annotation results. 

### 4.5. Identification of Transcription Factors, Alternative Splicing Events, and lncRNAs

Transcription factors were predicted using the iTAK software (version 1.7a) [78] based on the plant TF databases. The software SUPPA (version 2.3) [79] was used to analyze alternative splicing events from Iso-Seq data. AS events are classified into seven categories: alternative 5′ donor (A5), alternative 3′ donor (A3), alternative first exon (AF), alternative last exon (AL), retention intron (RI), mutually exclusive exons (MX) and exon skipping (SE) events. Transcripts longer than 200 nt with more than two exons were selected as lncRNA candidates. Four current mainstream coding potential analysis databases (PLEK [80], CNCI [81], CPC [82], and Pfam [83]) were combined to sort non-protein-coding RNA candidates from putative protein-coding RNAs in the transcripts. Take the intersection of transcripts without coding potential in these software results as the candidate lncRNA data and applied for the subsequent analysis. 

### 4.6. Coexpression Network Analysis of Se Metabolism-Related DEmRNAs and TFs, DElncRNAs

The R package weighted gene coexpression network analysis (WGCNA) (version 1.42) was used to identify the modules of highly correlated genes based on the normalized expression matrix data. The screened transcripts related to Se metabolism and TFs; differentially expressed lncRNAs were included. FPKM values of the transcripts from the 18 independent samples belonging to root and leaf tissues with three treatment groups were used for WGCNA. Modules were obtained using the automatic network construction function blockwise modules with the default settings. As the method described by Gerttula et al. [84], the module was associated with the trait (the content of Se species detected), and the correlation matrix between the module and the trait was calculated. The module with the highest correlation coefficient and the smallest *p*-value was designated as the module most relevant to the trait. This study identified a significantly correlated module based on a correlation coefficient (r) ≥ 0.9 and *p* < 0.01. The coexpression networks of lncRNAs and hub lncRNAs in highly correlated modules were generated with the Cytoscape software (version 3.7.1) [85].

### 4.7. Validation of RNA-seq Data by RT-qPCR

Five lncRNAs were randomly selected for RT-qPCR verification. The total RNA sample isolated above was used as the template and reverse transcribed using the Evo M-MLV RT-PCR Kit (Accurate Biology, China). The expression of the β-actin 3 gene was used as an internal control. RT-qPCR was performed with ChamQ™ Universal SYBR qPCR Master Mix (Vazyme, Nanjing, China) on a LightCycler 480 II58 device (Roche, Basel, Switzerland) according to the manufacturer’s protocol. The relative expression values were determined against the tissues using the comparative Ct method (2^−ΔΔCt^) and triplicate biological samples. The primers for RT-qPCR were designed using Primer3Plus (https://www.primer3plus.com, accessed on 15 October 2022), shown in Appendix A.

## 5. Conclusions

Overall, by combining NGS and SMRT technologies, we explored the transcriptomic changes in the roots and leaves of SeNPs-treated soybean seedlings and unveiled a comprehensive picture of the regulatory networks of SeNPs metabolism in soybean. A total of 117 transcripts, including 66 transporter genes, 45 assimilatory enzyme genes, and 6 specific genes, were identified to be putatively involved in SeNPs transport and biotransformation in soybean. DEGs analysis showed that the genes encoding transporters such as SULTR2;1, SULTR1;3, SULTR 3;1, NIP5;1, NIP6;1, PHO1 and Lsi2, and assimilation enzymes such as APS2, APR3, CYSK, CGS, METE1, CNIF3, MGL, SYM, MMT, were significantly differentially expressed by SeNPs treatments. Moreover, *SYM*, *APR3*, and *CNIF3*, which were predicted to be regulated by the 3 TFs and 9 lncRNAs, were found to be significantly correlated with contents of SeCys_2_, MeSeCys, Se^4+^, Se^6+^, and total Se, indicating these genes may play key roles in SeNPs accumulation and tolerance in soybean. This work deepens our understanding of the mechanism of Se metabolism in soybean and provides useful information for future research on engineering Se biofortification soybean plants.

## Figures and Tables

**Figure 1 plants-12-00789-f001:**
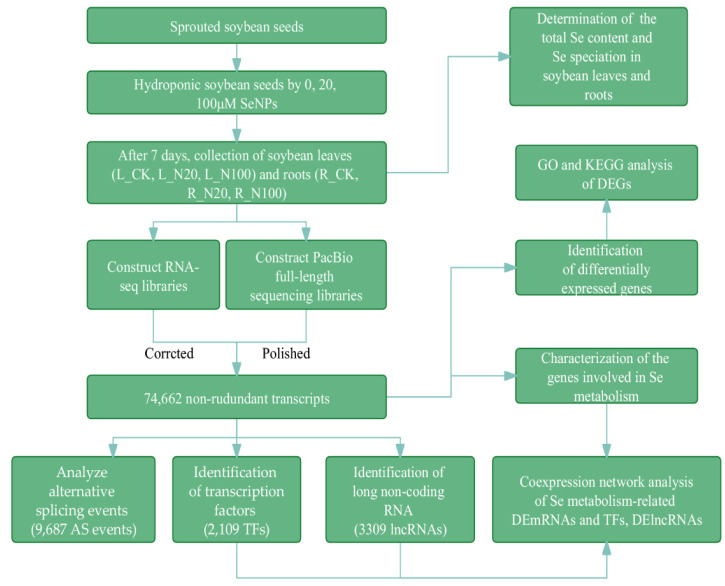
Overview of experimental and bioinformatics procedures in this study.

**Figure 2 plants-12-00789-f002:**
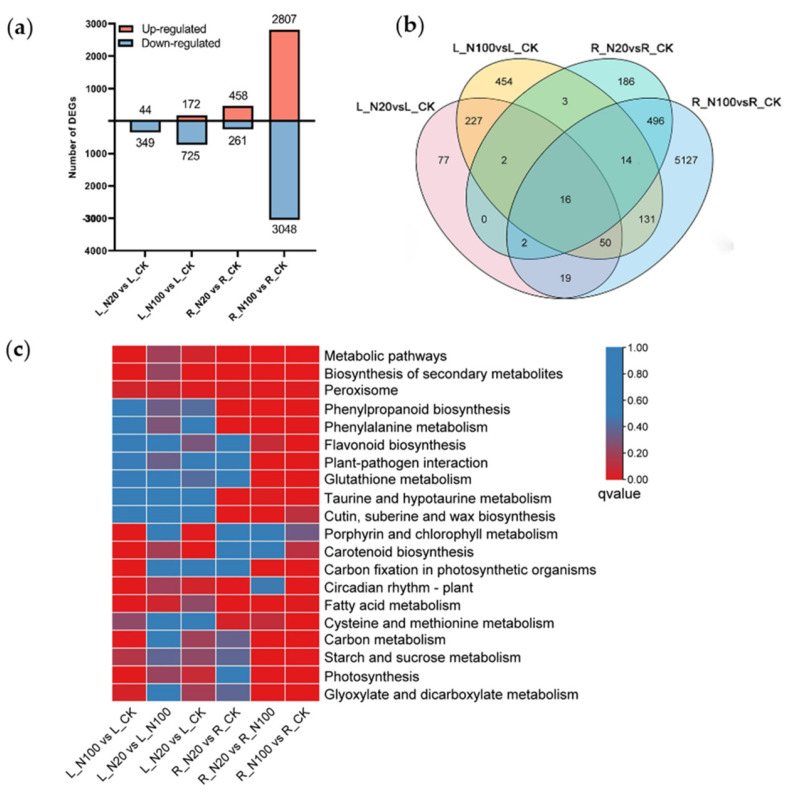
Number of DEGs and enrichment of KEGG pathways of the DEGs according to the *p*-value. (**a**) DEGs between the CK and treatment groups; (**b**) Venn of DEGs in different comparison groups; (**c**) Enrichment of the top 20 KEGG pathways of DEGs according to the q-value.

**Figure 3 plants-12-00789-f003:**
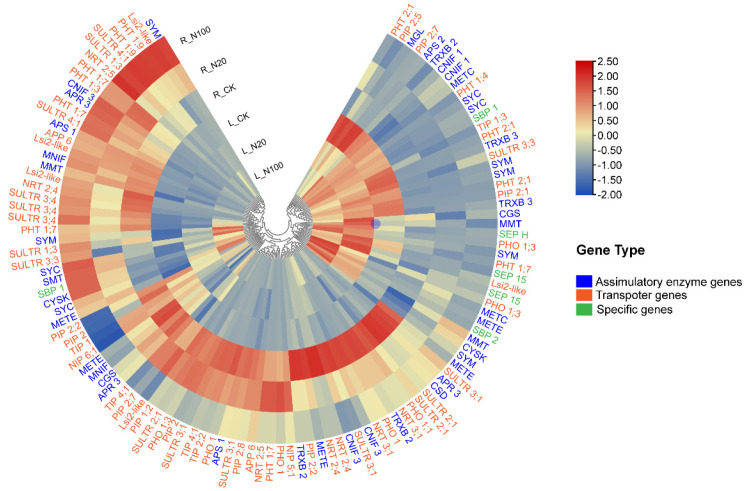
Global expression profile of the screened Se-metabolism-related transcripts. The transcripts were categorized into the transporter, assimilatory enzyme, and specific genes with different colors.

**Figure 4 plants-12-00789-f004:**
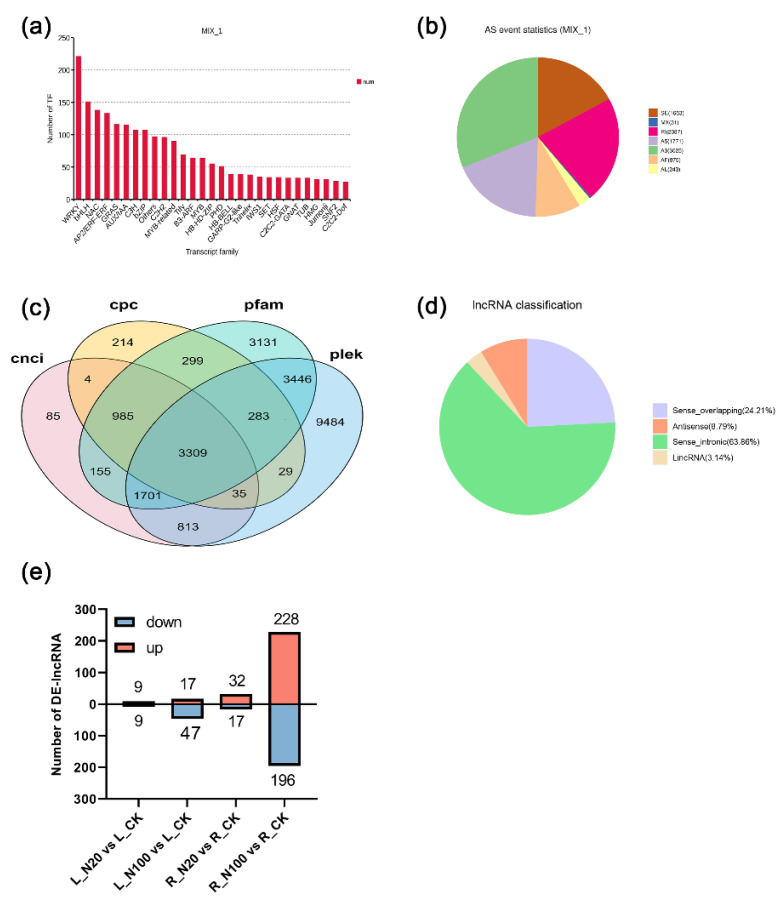
(**a**) TF family classification of genes; (**b**) Numbers of different types of alternative splicing events; (**c**) Venn diagram of lncRNA numbers from the four databases including CNCI, CPC, Pfam, and CLEK; (**d**) The classification of the lncRNA genes; (**e**) Number of differentially expressed lncRNA.

**Figure 5 plants-12-00789-f005:**
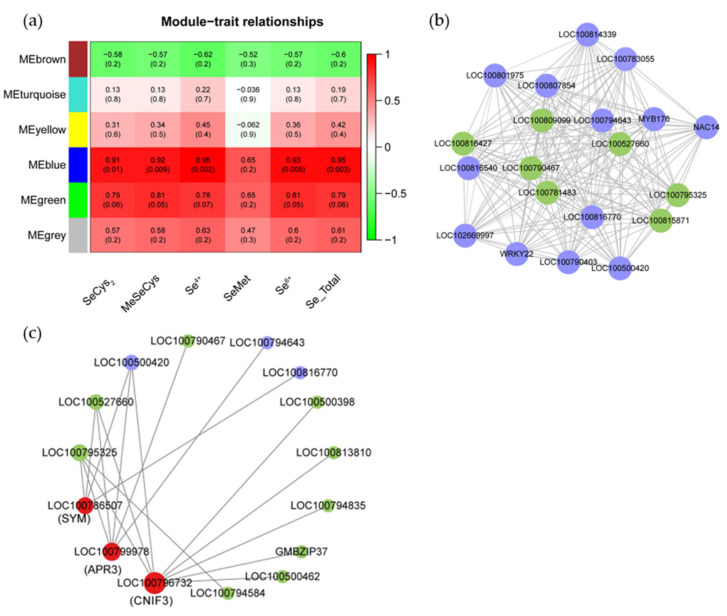
(**a**) Module-trait relationship analysis of TFs, lncRNAs, and transcripts involved in Se transport and assimilation genes. The same color module indicates the genes of the identical cluster. The numbers inside the boxes were Pearson’s correlation coefficient and their *p*-value in the brackets; (**b**) Coexpression network of DE-lncRNAs (green) and TFs (blue); (**c**) Interaction patterns between lncRNAs (green)/TFs (blue) and structural genes (red) related to Se metabolism.

**Figure 6 plants-12-00789-f006:**
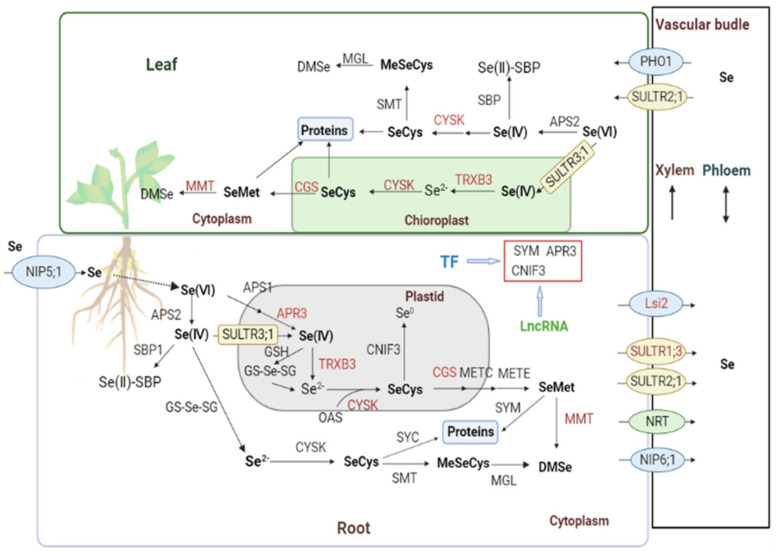
The putative SeNPs metabolism pathway in soybean. Enzymes in red indicate possible regulation by AS. Enzyme annotation: APS (ATP sulfurylase), APR (5′-adenylylsulfate reductase), TRXB (thioredoxin reductase), CYSK (cysteine synthase), SMT (selenocysteine methyltransferase), CNIF (cysteine desulfurase), CGS (cystathionine gamma-synthase), MMT (methionine S-methyltransferase), METC (cystathionine beta-lyase), METE (homocysteine methyltransferase), SYM (methionine-tRNA ligase), SYC (cysteine-tRNA ligase), MGL (methionine gamma-lyase) and SBP (selenium binding protein).

**Table 1 plants-12-00789-t001:** Contents (mg Se kg^−1^) and proportions*^a^* (%) of five Se species in soybean roots and leaves under different SeNPs treatments.

SOYBEAN PART	Treatment (μmol)	SeCys_2_	MeSeCys	Se^4+^	SeMet	Se^6+^	Se_Total
root	0 (L_CK)	ND	ND	0.03 ± 0.01 c (21.26 ± 4.97)	0.13 ± 0.02 c (78.74 ± 9.31)	ND	3.20 ± 0.73 c
	20 (L_N20)	2.25 ± 0.21 b (19.08 ± 1.82)	2.29 ± 0.29 b (19.48 ± 2.43)	0.89 ± 0.05 b (7.54 ± 4.33)	3.90 ± 0.24 b (33.12 ± 2.04)	2.44 ± 0.06 b (20.77 ± 0.48)	218.96 ± 22.24 b
	100 (L_N100)	6.32 ± 0.38 a (22.64 ± 0.13)	7.22 ± 0.34 a (25.89 ± 1.20)	1.76 ± 0.07 a (6.30 ± 0.24)	5.73 ± 0.25 a (20.55 ± 0.90)	6.87 ± 0.36 a (24.62 ± 1.28)	518.01 ± 29.82 a
leaf	0 (R_CK)	ND	ND	0.04 ± 0.01 c (15.35 ± 3.48)	0.19 ± 0.02 c (84.65 ± 9.13)	ND	1.45 ± 0.09 c
	20 (R_N20)	0.19 ± 0.02 b (6.09 ± 0.71)	0.19 ± 0.05 b (6.01 ± 1.56)	0.07 ± 0.02 b (2.30 ± 0.75)	2.18 ± 0.26 b (70.89 ± 0.84)	0.45 ± 0.00 b (14.71 ± 0.01)	15.47 ± 2.44 b
	100 (R_N100)	1.19 ± 0.20 a (17.34 ± 2.90)	0.95 ± 0.10 a (13.87 ± 1.41)	0.15 ± 0.03 a (2.14 ± 0.36)	3.79 ± 0.61 a (55.14 ± 8.89)	0.79 ± 0.09 a (11.52 ± 1.34)	42.85 ± 0.39 a

Data in brackets are proportions of Se species. ND, not detected; such data were not included in the analysis of variance. SeCys_2_, SeMeCys, and SeMet represent selenocystine, Se-methylselenocysteine, and selenomethionine, respectively. Results are presented as means ± SEs (*n* = 3). Different letters indicate significant differences in the content (or the proportion) of one of the Se species in shoots or roots, regardless of the exposure times (*p* < 0.05, LSD).

## Data Availability

Not applicable.

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
