# Peer review of "Illumina RNA and SMRT Sequencing Reveals the Mechanism of Uptake and Transformation of Selenium Nanoparticles in Soybean Seedlings"

_plants, 2023, doi:10.3390/plants12040789_

Round 1
Reviewer 1 Report
The topic of the article is novel, important, and of great importance. I will make some suggestions to improve the content of the manuscript.
1. Both in the Abstract and in the Results chapters, I found a description that says: "The KEGG enrichment analysis of these DEGs showed that biosynthesis of carotenoids, biosynthesis of secondary metabolites, porphyrin and chlorophyll metabolism were most significant in leaves , and enriched in secondary biosynthesis. metabolites, metabolic pathways, phenylpropanoid biosynthesis and phenylalanine metabolism were enriched roots after exposure to SeNP." This may be one of the most important findings of the authors, but there is no explanation for the results obtained. Please supplement the finding with plant physiological or biochemical explanations of the possible reasons. Support your results with relevant literature references by specifying.
2. A few sentences in the Introduction chapter should be transferred to the Results chapter. These are the following: "In this study, we combined single-molecule real-time (SMRT) sequencing and nextgeneration sequencing (NGS) technology to generate expression profiles of soybean roots and leaves under SeNPs exposure. In total, 74,662 non-redundant sequences were obtained from soybean roots and leaves, and differentially expressed genes (DEGs) were identified. A total of 117 transcripts including 66 transporter genes, 45 assimilatory enzyme genes, and 6 specific genes were identified to be putatively involved in the uptake, translocation, and biotransformation of SeNPs in soybean. Moreover, we identified 2109 transcription factors (TFs) and 491 differentially expressed lncRNAs (DE-lncRNAs), some of which were coexpressed with several Se metabolism-related genes and significantly correlated with contents of selenocysteine (SeCys2), MeSeCys, Se4+, Se6+, and total Se."
3. In the Results section, the authors state that: "y. In addition, the total Se contents in roots were much higher than that in leaves in the same treatment, indicating that Se was mainly stored in soybean roots after SeNPs uptake from the culture solution." There is no explanation for this statement anywhere in the article. Please explore possible reasons and support them with relevant references.
4. The soybean reference genome is referred to several times in the text: "Most of the polished consensus sequences (98.95%) were mapped to the reference genome (G. max v4.0) using GMAP software." and "Next, we compared the 42,215 corrected transcripts against the soybean genome set (G. max v4.0). " and "The error-corrected isoforms were mapped to the G. max v4.0 reference genome with GMAP software and then the redundant isoforms (identity > 0.9, coverage < 0.85) were eliminated using the ToFU package." Does the article not mention where the reference genome comes from? If it is from a public database, please include the official deposit number and access path in the text. If this is your own "de novo" constructed genome, please deposit the TSA in the official (NCBI) database and then refer to the accession number in the article. In the latter case, however, you cannot refer to it as a reference genome, the term "de novo" assembled soybean genome is more accurate.
5. "At the same time, more genes were down-regulated than up-regulated (Figure 1a)". No explanation was given for this statement in the article.
6. RT-qPCR complements and validates in silico studies well. The output files of the RT-qPCR tests used to prove the completion of the test are placed in the Supplementary files. If available, please place them in the Supplementary files.
7. The GO diagrams prepared for the KEGG analysis are accepted and show the obtained results well. When there is so much data, it is sometimes more spectacular, if the cloud diagram is used, the results are more emphasized and more spectacular. It is not necessary to replace it, just a suggestion for the future.
8. A simple graphic abstract can also improve the quality of the article. Not mandatory, just recommended.
Reviewer 2 Report
The manuscript plants-2194058 aimed to understand better the mechanisms of SeNPs transport and biotransformation in plants, therefore authors analyzed the expression profiles of soybean roots and leaves under SeNPs exposure using Illumina and SMRT sequencing.
Overall, the manuscript is clear, very thorough, methodologically appropriate, elaborated in sufficient detail, relevant to the field, and well-structured. To better interpret the results of the manuscript, I have some suggestions. After developing them, I recommend the ms. for publication.
Abstract
- I think that the abstract is too long, and more concise wording should be used. Please state the purpose of the manuscript and then only the main results should be presented.
Introduction
- Please explain what is considered: a low and high concentration of Se. In human terms, what can be known about the concentration, does it have a cytotoxic effect, and how much can accumulate in the plants?
Results
- Software versions should be provided everywhere.
- The names of the samples should be covered both here and in the methodology, because it is not clear to the reader which name refers to which sample. I would recommend a summary diagram of the experimental design that shows this as well as the treatments and their conditions.
- As the Venn diagram shows there are many individually (characteristically) expressed transcripts. It should be informative to describe what KEGG categories they belong to. I suggest briefly addressing these categories in the results as well.
- Please add information to the most common enriched pathways in which aligned DEGs were up and down-regulated, I suggest an additional supplementary table to present this information.
- The sample pairs provide information about what the treatment caused in the given plant organ. However, they do not inform about the Se dose dependence. I recommend introducing additional 2 sample pairs for this purpose (L_N20 vs L_N100 and R_N20 vs. R_N100). Furthermore, it is striking that the L_N100 vs. L_Ck and R_N100 vs.R_CK samples indicate stress-related KEGG pathways suggesting this dosage is very strong for the plant cell. just for that reason, I would recommend introducing the proposed 2 sample pairs
- The images in fig 1 should be taken sharper.
- Figure 2: Since the authors do not publish the complete annotation table of the total annotated transcripts, it would be much more informative to write the gene names or abbreviations instead of the identifiers. I would recommend a vertical heatmap instead of this figure if the authors want to include the cladogram as well because at the moment it is not interpretable
- The figures in fig 3 should be sharper.
- Figure4 a: Please indicate also in the figure caption what the color markings cover
Discussion
- Depending on the results obtained, I please introduce this section with a discussion summarizing the metabolic processes which were found affected by the treatments (uptake, transport, transformation, synthesis, toxicity, etc.)
Materials and Methods
- This section is sufficiently detailed. Please indicate sample nomenclature and the used software versions in the text.
- RT-qPCR: Please demonstrate the in silico gene expression index (fpkm) of the reference gene. Please discuss why this reference gene was chosen and convince the reader that its value does not change significantly after the treatments.
Reviewer 3 Report
The manuscript submitted by Xiaong et al. represent a significant progress in the topic and will be very interesting to a broad readers. The work was designed properly, data were analyzed completely and accurately, the manuscript was written well and clearly.
Minor comments:
Abstract: "9687 alternative splices" - it is not clear what it means - alternatively splice genes, or events, or isoforms?
Page 3: "among which microbial reduction from Se oxyanions is frequently" - should use "frequent".
Table 1: make the data format consistent, for example, keep two digits after decimal for all.
Figure 1 c-f: fonts too small. Suggest to use larger fonts.
Isomers on Page 8: I do not see this term used often in literature, isoforms are often used.
Page 14: Materials and methods, during germination and seedling growth, what is the light condition or totally in dark?
"seedlings .... and separated into roots and leaves" - for such a young seedling, hypocotyl tissue may account a large proportion, do you include hypocotyl materials in the "leaf" tissue or discarded?
Were the RNA-seq data deposited to a public database such as NCBI SRA database?
Round 2
Reviewer 1 Report
The authors have made the changes I suggested, except for point 6.
Point 6: RT-qPCR complements and validates in silico studies well. The output files of the RT-qPCR tests used to prove the completion of the test are placed in the Supplementary files. If available, please place them in the Supplementary files.
Response 6: Thank you very much for your suggestion, but I am very sorry that there is no output file here, we are getting the data as exported data, therefore no specific output file is saved.
However, because of this, they do not have to repeat expensive tests. I accept their answers, I recommend the corrected article for publication.